# The struggle for medicine: A valid and reliable cross-sectional study on the impact of war on healthcare access and its consequences for displaced Sudanese citizens

Ali Awadallah Saeed[1]*, Ahmad Mohammad Al Zamel[2], Aya Al-waleed Galal Ahmed[3], Dina Abdelrahman Abdelltif Abdelrahman[3], Eilaf Faisal Ataa Abdoon[3], Nuha Hussein Mohammed Mousa[3], Rayan Ibrahim Alrasheed Salim[3], Samah Mahgoub Ahmed Mahgoub[3], Shahd Amir Gamalaldeen Mohammed[3], Safaa Badi[4]

**1** Department of Pharmacology, Faculty of Clinical and Industrial Pharmacy, National University-Sudan, Mycetoma Research Center, Khartoum, Sudan, **2** Al-Neelain University, Faculty of Medicine and Surgery, Khartoum, Sudan, **3** Faculty of Clinical and Industrial Pharmacy, National University-Sudan, **4** Department of Clinical Pharmacy, Faculty of Pharmacy, Omdurman Islamic University, Khartoum, Sudan

* alimhsd@gmail.com

## Abstract

Armed conflicts severely impact healthcare systems leading to medication shortages and restricted access to essential services. The ongoing war in Sudan has disrupted healthcare infrastructure affecting patients particularly those with chronic diseases. This study examines the accessibility of medications and the consequences of limited healthcare access during the conflict. A descriptive cross-sectional study was conducted among individuals affected by the Sudan war. Data were collected through an online questionnaire assessing medication accessibility, healthcare service availability, and socioeconomic factors. Statistical analysis was performed using SPSS to examine correlations between accessibility and health outcomes. Out of 300 participants, 56.7% reported poor medication accessibility while 43.3% had to relocate due to a lack of medical care. 65.7% experienced worsening health condition and 61.3% believed medication shortages contributed to increased mortality. Low-income and unemployed individuals faced the greatest challenges in accessing medications. The Sudan war has significantly disrupted healthcare access with severe consequences for medication availability and patient health. Urgent humanitarian interventions and policy measures are needed to restore medication supply chains and improve healthcare access for conflict-affected populations.

## Introduction

Armed conflicts severely weaken national healthcare systems often resulting in the collapse of critical services. In Sudan, intense fighting has left over 11 million people

---

---

**Data availability statement:** All relevant data are within the paper.

**Funding:** The author(s) received no specific funding for this work.

**Competing interests:** The authors have declared that no competing interests exist.

in urgent need of medical care exacerbating pre-existing healthcare challenges [1,2]. Similarly, the war in Ukraine has led to significant declines in health service functionality, disrupting access to medications, emergency care, and chronic disease management [3,4]. Such disruptions are not unique; past conflicts in Yemen, Syria, and Ethiopia have demonstrated that healthcare systems in war zones become fragmented, with limited functionality and overwhelmed facilities [5–8]. One of the most pressing concerns in conflict-affected areas is the increase in preventable mortality and morbidity due to the breakdown of essential medical services. Studies show that during the Tigray war, chronic disease patients faced severe treatment interruptions leading to increased morbidity and mortality [9]. In Gaza, shortages of medicine, food, and water have left thousands unable to manage chronic conditions such as asthma, kidney disease, and diabetes [10]. Similarly, in Ukraine only 10% of pharmacies remained operational in the initial days of the war limiting access to essential medications for chronic disease management [3]. A common consequence of war is disruptions in drug supply chains making it difficult for populations to access essential medications. In Sudan, the conflict has led to widespread shortages of life-saving medicines particularly for chronic conditions like diabetes, hypertension, and kidney disease [11–13]. Studies on Ukraine and Yemen highlight similar challenges, where wartime instability has caused severe medication shortages leading patients to ration their doses or substitute medications often with serious health consequences [3,5]. The pharmaceutical industry is particularly vulnerable during armed conflicts [14]. Julian Upton analyzed the impact of war on global drug supply chains revealing that armed conflicts disrupt pharmaceutical production, transportation, and distribution networks further worsening medication shortages [15]. Similar findings by Shukar et al. (2021) indicate that war-induced drug shortages increase the risk of medication errors contributing to higher mortality rates [16]. Beyond healthcare access wars also create severe economic instability making it harder for patients to afford medical care. The collapse of employment rising inflation and loss of income leave many unable to purchase essential medicines even when available [17,18]. In Sudan, economic decline has exacerbated healthcare inequalities limiting access to life-saving treatments [19]. Forced displacement further worsens healthcare access as many patients are forced to relocate to areas with limited or no medical facilities [20]. Studies on Syria, Ukraine, Ethiopia and Sudan show that displaced individuals often struggle to access healthcare services leading to further deterioration in health outcomes [21,22]. This aligns with findings from Dzhus & Golovach who reported that the war in Ukraine led to large-scale displacement forcing patients to travel long distances in search of medical care often without success [23]. Despite extensive research on war-related healthcare disruptions in Ukraine, Syria, and Yemen, limited studies focus on medication accessibility during the Sudan war. Existing literature highlights infrastructure destruction and drug shortages but lacks quantitative insights on Sudanese patients. This study fills that gap by assessing medication access, health impacts, and socioeconomic factors, providing crucial data for policy and humanitarian interventions.

## Methodology

### Ethics statement

Ethical approval was obtained from the ethical committee of the department of clinical pharmacy; faculty of pharmacy, National university- Sudan under reference serial number was (FPNU-09–4/2023). The objectives of the study were clarified to the participants and written informed consent was obtained. Written informed consent was obtained from all participants prior to filling out the questionnaire, in accordance with the Declaration of Helsinki.

Study design: This study employed a descriptive cross sectional design conducted among citizens residing in or displaced from war-affected areas in Sudan.

Study period: The data collection period spanned from 04/10/2023 to 1/11/2023 lasting four weeks.

### Study area and population

The study covers four distinct areas in Sudan, which could contribute to regional diversity in responses.

The four authors were from these four areas, Al Kalakla district (A residential locality in Khartoum State), Marwai city (Northern State), Madani city (Gezira State), and Atbara city (Nile River State).

### Sample size

The sample size was calculated using Cochran's formula, which provides an ideal sample size given a desired level of precision, confidence level, and estimated population proportion: $N = (1.96)^2 (0.5)(0.5)(0.05)^2 = 385$

$$N = \frac{(1.96)^2(0.5)(0.5)}{(0.05)^2} = 385$$

$$N = (0.05)2(1.96)2(0.5)(0.5) = 385$$

Cochran's formula is widely used in survey research to calculate the sample size needed for estimating proportions in a population with a given level of confidence and precision [24].

### Study setting

The four authors, each based in different cities, employed **stratified random sampling** to ensure a representative sample by dividing the population into strata based on demographic factors such as age, gender, and education level.

A **structured questionnaire** was used as the primary data collection tool. It was distributed to a **randomly selected** sample within each stratum, ensuring that participants from different demographic backgrounds were included.

This sampling technique enhanced the study's reliability by minimizing selection bias and ensuring a balanced representation of various population subgroups across the different cities.

The inclusion criteria encompassed Sudanese families whose healthcare access had been adversely affected by the conflict either directly or indirectly. Individuals who were not residents in conflict-affected areas during or before the study period were excluded.

### Research instrument

The questionnaire was developed by the researchers based on the study objectives and a comprehensive literature review of similar studies [21–23,25–28]. To ensure face validity, it was pre-tested with a sample of 12 participants to assess clarity and relevance.

The questionnaire consisted of three domains; the first domain was demographic and general characteristics of participants, the second domain was medications' accessibility evaluation subscale, which assessed the availability and ease of obtaining medications. This subscale demonstrated good internal consistency, with a Cronbach's alpha of 0.741 and a mean inter-item correlation of 0.290. The third domain was medication accessibility consequences subscale, which examined the impact of accessibility on treatment adherence and health outcomes. This subscale also showed good internal consistency, with a Cronbach's alpha of 0.804 and a mean inter-item correlation of 0.456.

The questionnaire was initially designed in English and later translated into Arabic to ensure better comprehension among respondents.

### Statistical analysis

Data was entered into Microsoft Excel and analyzed using SPSS version 27.0. Missing values across variables ranged between 0% and 0.67%. Descriptive statistics (frequencies and percentages) were used to summarize categorical variables. Normality was assessed using the Kolmogorov-Smirnov and Shapiro-Wilk tests. Pearson's correlation was used to examine associations between medication accessibility and its consequences. A one-way ANOVA was conducted to evaluate the effects of income, employment status, and place of residence on medication accessibility with post hoc comparisons performed using Tukey's HSD test. A p-value of less than 0.05 was considered statistically significant.

## Results

### Demographic characteristics

A total of 300 participants completed the questionnaire with 59.0% being female and 41.0% male. The majority (54.3%) were under the age of 35 while 45.7% were 35 years or older. In terms of education, 76.5% had a college degree or higher whereas 18.4% had an education level below college. Before the war, 90.0% of participants resided in Khartoum. However, after the outbreak of conflict there was a ninefold increase in those relocating outside the capital with only 5.7% remaining in Khartoum during war. The study explored participants' health conditions revealing that the majority had no chronic diseases. Among those with pre-existing conditions 27.1% had diabetes, 17.9% had hypertension, and 16.4% had both diabetes and hypertension. Additionally, 19.3% reported suffering from asthma (Table 1).

### Health and economic factors

The war significantly impacted employment and financial stability with the number of unemployed participants rising from 17 to 150 participants. (Table 2) Similarly, income levels were severely affected as low-income individuals increased from 37 pre-war to 173 participants during war (Fig 1).

### Medication accessibility

The most widely reported issue was the disruption or discontinuation of health services with 59.7% of participants strongly agreeing that healthcare facilities in their area were no longer functional due to the war. Affordability was also a key concern with 50.0% of respondents indicating that high medication prices made access difficult. Additionally, 51.3% faced challenges reaching healthcare facilities due to fuel shortages and limited transportation.

PLOS Global Public Health

**Table 1. General characteristics of the study participants and clinical data (n = 300).**

| General characteristics | | Frequency (%) |
|---|---|---|
| Gender | Female | 177 (59%) |
| | Male | 123 (41%) |
| Age | Less than 35 | 163 (54.3%) |
| | 35 and more | 137 (45.7%) |
| Level of education | Master's degree or above | 46 (15.3%) |
| | Bachelor's degree | 181 (60.3%) |
| | (3) years Diploma | 3 (0.9%) |
| | High school diploma | 41 (13.7%) |
| | Less than High school diploma | 14 (4.7%) |
| | Prefer not to answer | 15 (5%) |
| Residence before war | In Khartoum | 270 (90%) |
| | Outside Khartoum | 30 (10%) |
| Residence after war | In Khartoum | 16 (5.7%) |
| | Outside Khartoum | 284 (94.3%) |
| Chronic diseases | Asthma | 58(19.3%) |
| | Diabetes | 81(27.1%) |
| | Hypertension | 54(17.9%) |
| | Diabetes and Hypertension | 49(16.4%) |
| | No | 58(19.3%) |

**Table 2. Distribution of the participants according to health and economic factors (n = 300).**

| Employment Status | Pre-war employment, N (%) | 3 months after war, N (%) | Current employment N (%) |
|---|---|---|---|
| Full-time | 49 (16.3) | 9 (3) | 16 (5.33) |
| Part-time | 39 (13) | 18 (6) | 35 (11.67) |
| Self-employed | 66 (22) | 33 (11) | 39 (13) |
| Students | 101 (33.67) | 48 (16) | 52 (17.33) |
| Retired | 4 (1.33) | 12 (4) | 10 (3.33) |
| Housewife | 24 (8) | 30 (10) | 34 (11.3) |
| Unemployed | 17 (5.7) | 150 (50) | 114 (38) |

Outside Khartoum, 55.7% of participants reported that pharmacies were running low on essential medicines. Medication accessibility was categorized into three levels as high, moderate, and poor. More than half of the participants (56.7%) reported poor accessibility while 36.3% had moderate access, and only 7.0% reported high accessibility (Table 3).

## Consequences of limited medication access

A significant proportion of participants (43.3%) reported that the unavailability of medication forced them to relocate. In addition, 42.7% resorted to using alternative drugs or adjusting their prescribed doses due to shortages and 33.3% reported reducing their medication intake. The most alarming findings were that 65.7% of respondents stated that their health deteriorated due to a lack of medication while 61.3% believed that medication shortages directly contributed to deaths (Table 4).

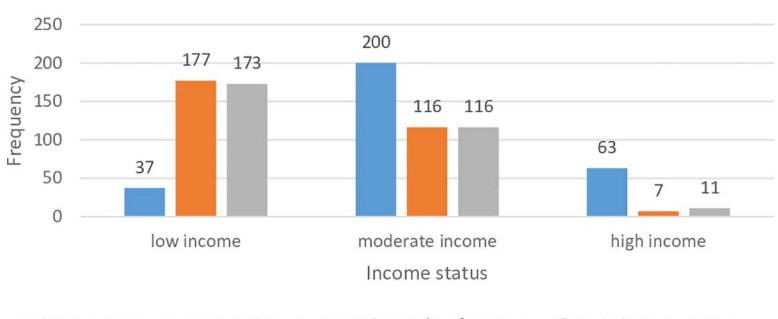

Participants' pre-war , three months after war and current income status

**Fig 1. Participants' Income Status: Pre-War, Three Months After War, and Current.**

**Table 3. articipants' opinions regarding medication's accessibility (n=300).**

| Accessibility factors | Strongly Disagree N % | Disagree N % | Neutral N % | Agree N % | Strongly Agree N % |
|---|---|---|---|---|---|
| You have difficulty getting medication or going to health care facilities | 13 4.3% | 27 9.0% | 51 17.0% | 72 (24.0%) | 137 (45.7%) |
| Some health services in your area have been disrupted or discontinued as a result of the war | 7 2.3% | 20 6.7% | 27 9.0% | 67 (22.3%) | 179 (59.7%) |
| you suffer from drug damage as a result of power outage | 16 5.3% | 38 12.7% | 39 13.0% | 71 23.7% | 136 (45.3%) |
| Medication is not readily available in your area | 41 13.7% | 41 13.7% | 82 27.3% | 69 23.0% | 67 (22.3%) |
| Medicines are difficult to get due to their high prices | 4 1.3% | 14 4.7% | 42 14.0% | 90 30.0% | 150 (50.0%) |
| You may find it difficult to get to health care facilities due to a lack of transportation and fuel | 9 3.0% | 20 6.7% | 42 14.0% | 75 25.0% | 154 (51.3%) |
| Outside of Khartoum, pharmacies are running low on medicine | 4 1.3% | 12 4.0% | 40 13.3% | 77 25.7% | 167 (55.7%) |
| Medications' accessibility | **Categories** | **N (%)** | | | |
| | High | 21(7%) | | | |
| | Moderate | 109(36.3%) | | | |
| | Poor | 170(56.7%) | | | |

**Table 4. Consequences of medication accessibility (n = 300).**

| The impact of medications' accessibility | *Strongly Disagree N (%)* | *Disagree N (%)* | *Neutral N (%)* | *Agree N (%)* | *Strongly Agree N (%)* |
|---|---|---|---|---|---|
| The unavailability of medication was a major reason to leave your place of residence | 24 (8.0%) | 34 (11.3%) | 59 (19.7%) | 53 (17.7%) | 130 (43.3%) |
| I had to use alternative drugs or alternative doses because of its unavailability | 13 (4.3%) | 33 (11.0%) | 51 (17.0%) | 75 (25.0%) | 128 (42.7%) |
| Due to a drug shortage, I had to lower my doses | 19 (6.3%) | 50 (16.7%) | 56 (18.7%) | 75 (25.0%) | 100 (33.3%) |
| Patients' health deteriorates as a result of being unable to get medication | 5 (1.7%) | 13 (4.3%) | 26 (8.7%) | 59 (19.7%) | 197 (65.7%) |
| Medication shortages led to deaths | 8 (2.7%) | 16 (5.3%) | 27 (9.0%) | 65 (21.7%) | 184 (61.3%) |

## Correlation between medication accessibility and health outcomes

Spearman correlation analysis examined the relationship between medication accessibility and its consequences. The results showed a strong positive correlation between poor accessibility and adverse health outcomes ($r > 0.50$, $p < 0.001$). Specifically, participants with poor access to medications were significantly more likely to relocate, adjust or lower their dosages, and experience health deterioration. The highest correlation was observed between medication inaccessibility and worsening health conditions ($r = +0.59$, $p < 0.001$) and between poor accessibility and increased mortality ($r = +0.60$, $p < 0.001$) (Table 5).

## Impact of socioeconomic factors on medication accessibility

A one-way ANOVA was conducted to evaluate the impact of income, employment status, and place of residence on medication accessibility. The results revealed significant differences based on income with low-income participants experiencing significantly greater difficulties in accessing medications compared to those with moderate income ($p = 0.020$). Employment status also played a significant role with students reporting significantly better medication access compared to unemployed individuals ($p < 0.001$). However, there was no statistically significant difference in medication accessibility between participants residing in Khartoum and those outside the capital ($p = 0.494$) (Table 6).

## Discussion

The destruction of healthcare facilities has been one of the most catastrophic consequences of the conflict in Sudan. This study found that over half of participants reported that health services in their area had been disrupted or discontinued due to the war. Similar findings have been observed in other war-affected regions. In Ukraine for example Haque et al.

**Table 5. Correlation between medication accessibility and health outcomes.**

| variables | | Spearman correlation | p-value |
|---|---|---|---|
| Total score of validated seven-point of Accessibility evaluations | The unavailability of medication was a major reason to leave your place of residence | + 0.55 | < 0.001 |
| | I had to use alternative drugs or alternative doses because of its unavailability | +0.50 | < 0.001 |
| | Due to a drug shortage, I had to lower my doses | + 0.55 | < 0.001 |
| | Patients' health deteriorates as a result of being unable to get medication | + 0.59 | < 0.001 |
| | Medication shortages led to deaths | + 0.60 | < 0.001 |

**Table 6. Analysis of variance for current income status, employment status and place of residence by medication accessibility score. (n = 300).**

| Dependent variable | Source of variation | | Mean value | SD | p-value |
|---|---|---|---|---|---|
| Total score of validated seven-point Accessibility evaluations | **Current income status** | Low income | 28.81 | 4.311 | 0.020 |
| | | Moderate income | 27.28 | 5.186 | |
| | | High income | 26.64 | 8.721 | |
| | **Current employment status** | Full-time employment | 27.75 | 6.224 | < 0.001 |
| | | Part-time employment | 30.91 | 3.416 | |
| | | Self-employed | 28.03 | 6.251 | |
| | | Student | 25.73 | 4.459 | |
| | | Retired | 27.90 | 3.315 | |
| | | Housewife | 27.26 | 4.085 | |
| | | Unemployed | 28.75 | 4.674 | |
| | **Current place of residence** | Khartoum | 29.00 | 4.243 | 0.494 |
| | | Outside Khartoum | 28.08 | 4.953 | |

[4] documented the Russia-Ukraine war led to the collapse of many hospital services disrupting patient care and limiting access to essential treatments. Likewise, Badri & Dawood [12] reported the Sudanese war has severely damaged hospitals resulting in significant shortages of medical staff, essential drugs, and critical medical supplies. War-related medication shortages are a global issue that affects both acute and chronic disease management. The current study found that 50% of participants faced difficulty accessing medications due to high costs and 55.7% reported that pharmacies outside Khartoum were running low on essential medicines. This aligns with findings from Yemen where a study by Mohamed Ibrahim et al. [26] demonstrated that ongoing conflict led to severe shortages of essential medicines including antibiotics, insulin, and cardiovascular drugs exacerbating mortality and morbidity rates among the affected population. Similarly, Khanyk et al. [3] found that the war in Ukraine severely disrupted long-term medication supply chains particularly for chronic disease patients who rely on continuous medication. The lack of transportation and fuel shortages further complicates medication accessibility. In this study, 51.3% of participants indicated difficulties in reaching healthcare facilities due to transportation issues which echoes findings from Ethiopia where war-related fuel shortages and roadblocks significantly reduced healthcare access for displaced populations [27].

Patients with chronic conditions such as diabetes, hypertension, and kidney disease face disproportionate risks during armed conflicts. This study found that a large proportion of participants with chronic diseases experienced worsening health conditions due to medication unavailability with 65.7% of respondents reporting health deterioration and 61.3% believing medication shortages contributed to mortality. These findings are supported by Zeleke et al. [28] who found that in Ethiopia adherence to chronic disease treatment significantly declined during war leading to severe complications and increased mortality. A particularly vulnerable group is patients requiring dialysis. In Sudan, ongoing conflict has left many dialysis patients without access to regular treatment. Konozy [22] reported that Sudan's dialysis centers have been overwhelmed with a 183% increase in patient load leading to reduced treatment frequency and an increased risk of fatal complications. A similar crisis was observed in Ukraine where Sulaieva et al. [29] found that war disrupted insulin supply for diabetic patients leading to preventable complications and deaths.

The economic impact of war has been profound leading to job losses, income reductions, and increased financial burdens on healthcare access. This study found that low-income individuals faced significantly greater difficulties in accessing medications compared to those with moderate incomes. This aligns with findings by Mohamed & Lucero-Prisno [30] who emphasized that Sudan's economic collapse due to war has exacerbated healthcare inequalities pushing many citizens into extreme poverty and limiting their ability to afford necessary treatments. Additionally, 43.3% of participants in this study reported relocating due to medication shortages highlighting the role of forced displacement in worsening health outcomes. Similar patterns have been observed in Syria where war-induced displacement significantly disrupted healthcare access particularly for chronic disease patients [31]. Dzhus & Golovach [23] also reported that the war in Ukraine led to large-scale displacement forcing patients to travel long distances in search of medical care often unsuccessfully.

The findings of this study align with broader global trends in conflict-affected regions. The World Health Organization (WHO) [32] reported that access to healthcare services in war zones is becoming increasingly difficult worldwide with essential medicines and services often unaffordable or inaccessible. This is evident in multiple conflicts including Sudan, Syria, Yemen, and Ukraine where healthcare systems have been pushed to the brink of collapse. Druce et al. [33] emphasized the need for robust strategies to protect and maintain healthcare services during armed conflict including mobile clinics, strengthened humanitarian supply chains, and international policy interventions. The pharmaceutical supply chain is particularly vulnerable in war settings. Julian Upton [15] analyzed the impact of war on global drug supply chains noting that armed conflicts lead to severe disruptions in pharmaceutical production and distribution further exacerbating medication shortages. Similarly, Shukar et al. [18] highlighted that drug shortages increase the risk of medication errors leading to higher morbidity and mortality rates among patients in conflict zones. Given the severity of the healthcare crisis in Sudan urgent interventions are needed to address medication shortages, improve healthcare infrastructure, and support

displaced populations. Druce et al. [33] outlined several key approaches to maintaining healthcare services in conflict zones including:

- Enhancing the coordination of humanitarian assistance to ensure the delivery of essential medications and supplies to impacted populations.Implementing mobile and remote healthcare services to sustain medical assistance in high-conflict areas.

- Improving pharmaceutical inventory and distribution systems to mitigate the effects of supply chain disruptions.

Moreover, international organizations ought to prioritize healthcare impartiality in conflict zones. Ewelike et al. [34] emphasize in their study of the Gaza war that safeguarding healthcare workers and facilities from targeted assaults is essential for maintaining medical services during conflict.

## Limitations

This study captures a single point in time and may not reflect long-term trends. The sample may not include the most vulnerable populations and responses may have some bias. Despite this, the findings highlight important challenges in accessing medication during the Sudan war.

## Conclusion

The war in Sudan has had devastating consequences on healthcare access and medication availability. Disproportionately affecting patients with chronic diseases. The findings of this study confirm global patterns observed in other war-affected regions where healthcare infrastructure damage, medication shortages, and economic instability collectively worsen health outcomes. Given the strong correlation between limited medication accessibility and increased morbidity and mortality, urgent action is needed to implement policies that ensure continued access to essential healthcare services. International cooperation, humanitarian interventions, and local policy reforms are critical to mitigating the crisis and protecting vulnerable populations.

## Author contributions

**Conceptualization:** Ali Awadallah Saeed, Ahmad Mohammad Al Zamel, Aya Al-waleed Galal Ahmed, Dina Abdelrahman Abdelltif Abdelrahman, Eilaf Faisal Ataa Abdoon, Nuha Hussein Mohammed Mousa, Rayan Ibrahim Alrasheed Salim, Shahd Amir Gamalaldeen Mohammed.

**Data curation:** Ali Awadallah Saeed, Aya Al-waleed Galal Ahmed, Dina Abdelrahman Abdelltif Abdelrahman, Eilaf Faisal Ataa Abdoon, Nuha Hussein Mohammed Mousa, Rayan Ibrahim Alrasheed Salim, Samah Mahgoub Ahmed Mahgoub, Shahd Amir Gamalaldeen Mohammed, Safaa Badi.

**Formal analysis:** Safaa Badi.

**Investigation:** Dina Abdelrahman Abdelltif Abdelrahman, Eilaf Faisal Ataa Abdoon, Nuha Hussein Mohammed Mousa, Rayan Ibrahim Alrasheed Salim, Samah Mahgoub Ahmed Mahgoub, Shahd Amir Gamalaldeen Mohammed.

**Methodology:** Ali Awadallah Saeed, Dina Abdelrahman Abdelltif Abdelrahman.

**Resources:** Safaa Badi.

**Supervision:** Ali Awadallah Saeed, Dina Abdelrahman Abdelltif Abdelrahman.

**Validation:** Eilaf Faisal Ataa Abdoon, Nuha Hussein Mohammed Mousa, Rayan Ibrahim Alrasheed Salim, Shahd Amir Gamalaldeen Mohammed.

**Visualization:** Ahmad Mohammad Al Zamel, Eilaf Faisal Ataa Abdoon, Rayan Ibrahim Alrasheed Salim, Samah Mahgoub Ahmed Mahgoub.

**Writing – original draft:** Ali Awadallah Saeed, Ahmad Mohammad Al Zamel, Aya Al-waleed Galal Ahmed.

**Writing – review & editing:** Ali Awadallah Saeed, Aya Al-waleed Galal Ahmed.

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
