## [Decision Letter · Decision Letter 0]

4 Jun 2025

PGPH-D-25-00510

The Struggle for Medicine: A Valid and Reliable Cross-Sectional Study on the Impact of War on Healthcare Access and Its Consequences for Displaced Sudanese Citizens

Dear Dr. Ali Awadallah Saeed

Thank you for submitting your manuscript to PLOS Global Public Health. After careful consideration, we feel that it has merit but does not fully meet PLOS Global Public Health’s publication criteria as it currently stands. Therefore, we invite you to submit a revised version of the manuscript that addresses the points raised during the review process.

Please submit your revised manuscript by June 18 If you will need more time than this to complete your revisions, please reply to this message or contact the journal office at globalpubhealth@plos.org. Please include the following items when submitting your revised manuscript:

We look forward to receiving your revised manuscript.

Kind regards,

Nazik Hammad, MD, FACP, FRCPC

Academic Editor

Journal Requirements:

1. Please insert an Ethics Statement at the beginning of your Methods section, under a subheading 'Ethics Statement'.

2. We note that your Data Availability Statement is currently as follows: “All relevant data are within the paper.”.

Additional Editor Comments (if provided):

Please address each reviewer's comments and suggestions

Reviewers' comments:

Reviewer's Responses to Questions

**Comments to the Author**

1. Does this manuscript meet PLOS Global Public Health’s publication criteria?

Reviewer #1: Partly

Reviewer #2: Yes

2. Has the statistical analysis been performed appropriately and rigorously?

Reviewer #1: Yes

Reviewer #2: Yes

3. Have the authors made all data underlying the findings in their manuscript fully available (please refer to the Data Availability Statement at the start of the manuscript PDF file)?

Reviewer #1: No

Reviewer #2: Yes

4. Is the manuscript presented in an intelligible fashion and written in standard English?

Reviewer #1: No

Reviewer #2: Yes

Reviewer #1: This manuscript addresses a critically important and timely issue: the disruption of healthcare access during the ongoing conflict in Sudan, with a specific focus on medication availability for displaced and affected populations. The topic is highly relevant to global public health, and the authors should be commended for conducting research in such a challenging context.

However, several major concerns must be addressed before the manuscript can be considered for publication:

1. Data Collection Context

Data collection occurred between October 4 and November 1, 2023, a period of significant insecurity in Sudan. Conducting consistent and safe data collection via online methods during this time appears ambitious. The authors should provide more detail on how participants were reached, and the feasibility and safety of data collection in areas experiencing active conflict, especially among internally displaced populations (IDPs).

2. Sampling Strategy and Online Survey Bias

The feasibility of the stratified sampling method is unclear given the use of an online, self-administered survey. This approach likely excludes the most vulnerable groups (e.g., IDPs, people in rural areas without internet access, those with limited literacy), introducing selection bias and compromising representativeness. The authors should provide more information on the recruitment strategy, how stratification was operationalized, and a clear justification for using online data collection in a context where electricity and internet access may been severely compromised.

3. Sample Size and Power

The sample size calculation based on Cochran’s formula indicated a minimum of 385 participants, yet only 300 were included. The authors should conduct a post hoc power analysis to assess whether the final sample size is adequate to detect the reported correlations or group differences.

4. Sample Skewness

The sample is heavily skewed toward highly educated individuals (over 75% with university degrees), which is unlikely to reflect the broader Sudanese population, especially during conflict. The implications of this sampling bias should be explicitly discussed in the limitations.

5. Terminology Use of “Post-War”

The use of the term “post-war” is misleading, as the war in Sudan is ongoing. The authors should revise this terminology and clearly contextualize the data collection period within the conflict timeline.

6. Data Availability

Only summary statistics are presented in the manuscript. This does not meet PLOS's data availability requirements. The authors should upload a de-identified dataset as supplementary material or to a public repository. If there are legal or ethical restrictions, these should be clearly explained per PLOS’s exemption policy.

7. Ethical Considerations

While the manuscript mentions ethics approval and informed consent, important details are missing. The authors should clarify how informed consent was obtained, how confidentiality and participant safety were ensured, and provide the ethics approval reference number for transparency.

8. Generalizability of Findings

Given the limitations of the sampling method, the findings cannot be generalized to the broader displaced or conflict-affected population in Sudan. The authors should qualify their conclusions accordingly and discuss the influence of digital exclusion on the study results.

Moreover, the manuscript appears to imply that difficulties accessing medications may have led to relocation. However, in the context of war and political insecurity, relocation is far more likely driven by security threats than by barriers to pharmaceutical access. The authors should avoid overstating or misattributing causality unless relocation motives were directly measured. A clearer framing would treat healthcare access as a consequence, not a cause, of displacement.

9. Language and Presentation

The manuscript would benefit from English language editing to improve clarity, grammar, and overall readability. Several sentences could be more concise, and transitions between sections could be smoother.

Reviewer #2: Review: The struggle for medicine: a valid and reliable cross-sectional study on the impact of war on Healthcare access and its consequences for displaced Sudanese citizens

Dear PLOS Global Public Health team

Thank you for asking me to review this important paper which I would suggest publishing with a few minor changes as detailed below. It details previously unpubished data in this setting.

The key results this original study presents are the output from a questionnaire posed to those living in Sudan prior to (and in some cases during) the recent conflict and the effects that have been had on access to medication, health care services. This is significant as nothing of this kind has been published during the recent war in Sudan and it may help inform policy makers and healthcare providers to improve the situation in this conflict and in others. The authors have conducted a literature search relating to this issue in other areas of conflict.

Data and methodology

The data was derived from a questionnaire with good detail on how this was achieved to gain representative answers.

Results

I have checked the results from the figures and text match and note these discrepancies:

-line 175: those remaining in Khartoum-text says 4.7% and Table 1 says 5.7%

-line 177: those with hypertension-text says 19.3%, Table 1 says 17.9%

-line 178: those with hypertension and diabetes, text says 17.9%, Table 1 says 16.4%

-line 186-can I ask where these figures are from in the table (Table 2)-sorry if I’m missing it!

I’d be interested to know what diseases the majority of respondents had (line 176) if they weren’t chronic diseases. Can this be displayed?

Tables and Figures

All numbered and referenced in the text accurately

I have checked that figures in the text match those in the tables

A few comments:

-would a map of Sudan be useful with the areas sampled highlighted. This may show the uninformed reader the geography

Appropriate use of statistics and uncertainties

I am not a statistician, however the tests described appear appropriate for this setting. Uncertainties and limitations of this study clearly described.

Validity and conclusions

Conclusions drawn accurately reflect data presented with a well-informed discussion of results preceeding this

References

All checked and refer appropriately to what is referenced in the text

References are accurate and many recent

In the introduction, (line 101) I think the reference quoted as “30” should be “25”.

In order to keep the references continuous, line 129 should ideally be number 26 (Cochran’s formula)

Clarity and context

Main messages are well organised and made very clear

Suggested improvements

Line 118 “four authors were been”…:I would suggest rewording this sentence to be clearer, along the lines of “The four authors were from these four areas:”

Thank you again for inviting me to review this interesting and worthwhile paper.

Best wishes

**Do you want your identity to be public for this peer review?** For information about this choice, including consent withdrawal, please see our Privacy Policy

Reviewer #1: **Yes: ** Samiratou Ouedraogo

Reviewer #2: No

---

## [Editor Report · Decision Letter 1]

9 Nov 2025

The Struggle for Medicine: A Valid and Reliable Cross-Sectional Study on the Impact of War on Healthcare Access and Its Consequences for Displaced Sudanese Citizens

PGPH-D-25-00510R1

Dear Dr. Saeed,

We are pleased to inform you that your manuscript 'The Struggle for Medicine: A Valid and Reliable Cross-Sectional Study on the Impact of War on Healthcare Access and Its Consequences for Displaced Sudanese Citizens' has been provisionally accepted for publication in PLOS Global Public Health.

Best regards,

Julia Robinson

Executive Editor